# Quantitative CT at Follow-Up of COVID-19 Pneumonia: Relationship with Pulmonary Function Tests

**DOI:** 10.3390/diagnostics13213328

**Published:** 2023-10-27

**Authors:** Davide Colombi, Marcello Petrini, Camilla Risoli, Angelo Mangia, Gianluca Milanese, Mario Silva, Cosimo Franco, Nicola Sverzellati, Emanuele Michieletti

**Affiliations:** 1Radiology Unit, Department of Radiological Functions, AUSL Piacenza, Via Taverna 49, 29121 Piacenza, Italy; m.petrini@ausl.pc.it (M.P.); c.risoli@ausl.pc.it (C.R.);; 2Pulmonology Unit, Department of Emergency, AUSL Piacenza, Via Taverna 49, 29121 Piacenza, Italy; a.mangia@ausl.pc.it (A.M.); c.franco@ausl.pc.it (C.F.); 3Scienze Radiologiche, Department of Medicine and Surgery (DiMeC), University of Parma, Via Gramsci 14, 43126 Parma, Italy; gianluca.milanese@unipr.it (G.M.); mario.silva@unipr.it (M.S.);

**Keywords:** COVID-19, post-acute COVID-19 syndrome, pulmonary emphysema

## Abstract

Background: The role of quantitative chest computed tomography (CT) is controversial in the follow-up of patients with COVID-19 pneumonia. The aim of this study was to test during the follow-up of COVID-19 pneumonia the association between pulmonary function tests (PFTs) and quantitative parameters extrapolated from follow-up (FU) CT scans performed at least 6 months after COVID-19 onset. Methods: The study included patients older than 18 years old, admitted to the emergency department of our institution between 29 February 2020 and 31 December 2020, with a diagnosis of COVID-19 pneumonia, who underwent chest CT at admission and FU CT at least 6 months later; PFTs were performed within 6 months of FU CT. At FU CT, quantitative parameters of well-aerated lung and pneumonia extent were identified both visually and by software using CT density thresholds. The association between PFTs and quantitative parameters was tested by the calculation of the Spearman’s coefficient of rank correlation (rho). Results: The study included 40 patients (38% females; median age 63 years old, IQR, 56–71 years old). A significant correlation was identified between low attenuation areas% (%LAAs) <950 Hounsfield units (HU) and both forced expiratory volume in 1s/forced vital capacity (FEV1/FVC) ratio (rho −0.410, 95% CIs −0.639–−0.112, *p* = 0.008) and %DLCO (rho −0.426, 95% CIs −0.678–−0.084, *p* = 0.017). The remaining quantitative parameters failed to demonstrate a significant association with PFTs (*p* > 0.05). Conclusions: At follow-up, CT scans performed at least 6 months after COVID-19 pneumonia onset showed %LAAs that were inversely associated with %DLCO and could be considered a marker of irreversible lung damage.

## 1. Introduction

After a few months since the onset of the SARS-CoV-2 (COVID-19) pandemic in late 2019, clinical attention has focused on the long-term outcomes of patients with persistent or recurrent symptoms. Some patients continued to suffer from dyspnea, fatigue, exercise and activity limitation, brain fog, myalgia (muscle pain), and headaches, termed post-COVID-19 conditions, colloquially referred to as long COVID [1]. Paradoxically, long COVID is not related to COVID-19 severity, and frequently it develops in patients with normal or mild abnormal pulmonary function tests (PFTs) or chest CT [2]. Currently, the European Respiratory Society (ERS) and the European Society of Thoracic Imaging (ESTI) suggest performing chest CT follow-up at 3 months in severe/critical cases and longer only in patients with not resolved imaging or presenting with new/progressive respiratory symptoms [3]. The majority of the patients affected by COVID-19 pneumonia showed full resolution of the abnormalities identified at chest computed tomography (CT) scans [4,5]. Nevertheless, in around 40% of the patients, interstitial lung abnormalities (ILAs) were observed at 2-year follow-up (FU) CT scans [5]. Lung abnormalities at FU CT are stable 6 months after COVID-19 pneumonia onset [5]. Several studies demonstrated a not negligible rate (around 30–40%) of patients with altered diffusing capacity of lung for carbon monoxide (DLCO), lower than 75% of the predicted value, during follow-up after COVID-19 pneumonia [5,6]. Although CT is an imperfect test to assess lung function, a relationship between DLCO impairment and the visual assessment of post-COVID pneumonia abnormalities has been demonstrated [5]. Other modalities, such as dual energy CT or xenon 129 (^129^Xe) magnetic resonance (MR), started to offer insights into the pathophysiologic characteristics of this illness and showed a relationship with pulmonary function [7,8]. Nonetheless, these techniques are expensive and not widely available.

Quantitative CT is a useful prognostic tool for COVID-19 pneumonia at baseline CT [9,10]. Both the software quantification of the well-aerated lung (WAL) and of the high attenuation areas (HAAs) at chest CT are associated with a worse prognosis [9,10]. The role of quantitative CT at follow-up for COVID-19 pneumonia as a surrogate of lung function is controversial. Barini et al. failed to identify a significant relationship between PFTs and quantitative CT parameters at follow-up, such as healthy parenchyma, ground-glass opacities, consolidations, and pulmonary volume; nevertheless, low attenuation areas (LAAs) were not quantified [11]. Celik et al. demonstrated that LAAs were significantly higher at follow-up CT performed >30 days later than COVID-19 pneumonia diagnosis, speculating that emphysematous changes could be a marker of irreversible damage due to pulmonary inflammation caused by COVID-19 infection [12]. Thus, a comprehensive quantification of lung parenchyma at follow-up CT of COVID-19 pneumonia could identify parameters associated with PFTs. 

Therefore, the aim of the present study was to test the association between PFTs and quantitative parameters assessed visually or by software at FU CT scans at least 6 months after COVID-19 pneumonia onset. 

## 2. Materials and Methods

### 2.1. Study Population

This retrospective study was approved by the local ethics committee (Area Vasta Emilia-Nord) of our institution (institutional review board approval number: 241/2020/OSS/AUSLPC). Informed consent was obtained from all subjects involved in the study. The study included 83 patients older than 18 years old admitted to the emergency department (ED) of our institution between 29 February 2020 and 31 December 2020, with a positive nasal-pharyngeal swab for SARS-CoV-2, a chest CT scan (baseline CT) typical for COVID-19 at admission, and who underwent FU CT at least 6 months later. FU CT was performed in patients with a more severe clinical disease course or in patients presenting with new or progressive respiratory symptoms in the mid-long term after acute COVID-19 pneumonia [13]. Exclusion criteria (Figure 1) were 1. new infiltrate appearance at FU CT (*n* = 11), 2. severe CT artifacts (*n* = 2), 3. failure of quantitative CT analysis by software (*n* = 6), and 4. unavailable PFTs within 6 months from FU CT (*n* = 24). Ultimately, 40 patients were included in the study. The medical records of all patients were reviewed by two physicians (D.C. and A.M.). In particular, comorbidities, smoking history, and both breath rate (breaths/minute) and peripheral oxygen saturation (SpO2%) at ED admission were recorded. COVID-19 pneumonia was classified as severe when patients manifested at admission a respiratory rate greater than 30 breaths\minute, an SpO2% ≤ 93% in breathing room air, and acute respiratory distress syndrome (ARDS) was diagnosed when the ratio of arterial partial pressure of oxygen to fraction of inspired oxygen was 300 mm Hg or less [14]. In addition, oro-tracheal intubation or treatment with continuous positive airway pressure (CPAP) during the hospital stay was recorded for each patient. 

Spirometry at follow-up was performed using a flow-sensing spirometer and a body plethysmograph connected to a computer for data analysis (MasterScreen Body/Diffusion, CareFusion, San Diego, CA, USA). Forced vital capacity (FVC) and forced expiratory volume in 1 s (FEV1), expressed as an absolute value (liters) and as a percentage of the predicted value, were recorded. The FEV1/FVC ratio was also calculated. In addition, %DLCO was measured by the single-breath method. At least three measures were taken for every variable, with the purpose of guaranteeing the reproducibility of the data. %DLCO was regarded as abnormal when it was less than 75% of the predicted value [5]. The interpretation of the PFTs was based on the American Thoracic Society (ATS) and ERS guidelines at the time of patient selection [15]. 

### 2.2. Chest CT Acquisition and Interpretation

Baseline unenhanced CT scans were performed in the supine position during an inspiratory breath hold, moving from the apex to the lung bases, with a 16-slice scanner (Emotion 16; Siemens, Forchheim, Germany). FU unenhanced CT scans were obtained with either a 64-row CT scanner (Aquilon; Toshiba Inc., Tokyo, Japan) or two different 16-row CT scanners (Emotion 16, Siemens, and Brilliance 16, Philips Healthsystems, Amsterdam, The Netherlands). At FU CT, the technicians trained the patients to obtain a deep full inspiration with several practice breathing maneuvers demonstrated prior to obtaining scans. Both baseline and FU CT scans were performed with low-dose parameters as follows: tube voltage, 110 kV if body weight was 80 kg or less and 130 kV if body weight was higher than 80 kg; tube current, 40 mAs; pitch, one; collimation, 0.625 mm. Imaging datasets were reconstructed using a sharp kernel (FC 86 for Toshiba scanners, B70 for Siemens scanners, and LungB for Philips scanners) at 1–1.5 mm slice thickness with standard lung window settings (window width: 1500 HU and window center: −500 HU).

Visual assessment of both baseline and FU CT scans was performed independently by two radiologists (D.C. and M.P., with 8 and 6 years of experience, respectively), who were blinded to clinical data [9]. In both CT scans, the extent of COVID-19 pneumonia and WAL were visually assessed, as previously described [9,10]. Briefly, three lung zones were identified, namely, the upper zone above the level of the carina, the lower zone below the level of the intrapulmonary vein, and the middle zone between the upper and lower zones. For each zone, COVID-19 pneumonia and WAL extent were expressed as percentages of total lung volume and estimated to the nearest 5%; the scores were averaged to yield a global percentage. The consensus formulation for the visual scores was obtained as reported in the study by Cottin et al. [16]. At baseline CT, the COVID-19 pneumonia pattern was defined as predominant ground glass, predominant consolidations, or mixed ground glass and consolidations, as defined by the Fleischner Society Glossary of Terms for Thoracic Imaging [17]. Additionally, at FU CT, the appearance of bronchiectasis, architectural distortion, or honeycombing was evaluated [17]. At CT, the presence of emphysema was also detected, as defined by Hansell et al. [17]. The 5% most divergent observations for quantitative CT parameters and instances of discordance over the categorical CT assessment were resolved by consensus. 

Both baseline and FU CT images reconstructed with sharp kernels were anonymized and transferred to a dedicated workstation. A technician (C.R.) with seven years of experience obtained quantitative CT parameters by using commercially available software (IntelliSpace Portal, version 12.1; Philips Health System, Best, The Netherlands). The chronic obstructive pulmonary disease (COPD) application automatically segmented the whole lung and the airways, quantifying total lung volume (L). Later, after applying a noise-reduction algorithm, quantitative CT parameters as absolute values and as percentages of the total lung volume included between density threshold values defined manually were calculated. In particular, the percentage of low attenuation areas (%LAAs) as lung volume >−950 Hounsfield units (HU), the percentage of WAL (%S-WAL) as lung volume included between −950 HU and −750 HU, and the percentage of modified high attenuation area (%m-HAA) as lung volume >−750 HU were recorded. We used thresholds between −950 HU and −750 HU for the quantification of %WAL since the best correlation with the WAL visual score had been previously demonstrated [18]. When %LAAs were ≥5%, it was considered indicative of emphysema [19]. 

### 2.3. Statistical Analysis

Categorical and continuous variables were expressed as counts and percentages or medians with corresponding interquartile ranges (IQR). Differences in quantitative CT parameters between baseline and FU CT were tested using the Wilcoxon test for paired samples. The agreement between readers for visual and software-based quantitative CT parameters was assessed by the calculation of the intraclass correlation coefficient (ICC) for continuous variables or by the calculation of the Cohen’s weighted kappa (Kw) for categorical variables [20]. The interpretation of the ICC and the Kw was based on the following scale: <0.40 for poor agreement, 0.4–0.54 for weak agreement, 0.55–0.69 for moderate agreement, 0.70–0.84 for good agreement, and 0.85–1.00 for excellent agreement [21]. The correlation between quantitative CT parameters obtained from FU CT and PFTs was estimated by the calculation of the Spearman’s coefficient of rank correlation (rho) with the corresponding 95% confidence intervals (95% CI). A *p* value < 0.05 was considered significant. All data were recorded using a dedicated database (Excel 2010, Microsoft Corp., Redmond, WA, USA), and statistical analysis was performed using MedCalc software (version 14.8.1, MedCalc Software Ltd., Ostend, Belgium).

## 3. Results

### 3.1. Patient Characteristics and PFTs at Follow-Up

The study included 40 patients; 15/40 (38%) were females with a median age of 63 years old (IQR, 56–71 years old). Table 1 summarizes the main patients’ characteristics. Six out of 40 patients (15%) were current or former smokers. Pulmonary comorbidities were identified in 7/40 (17%) patients; the most frequent was COPD (3/40 patients, 7%). Besides pulmonary comorbidities, the majority of the patients were affected by systemic hypertension (18/40 of the patients, 45%). At ED admission, the median respiratory rate was 20 breaths/min (IQR, 18–25 breaths/min), while the median SpO2% was 92% (IQR, 89–96%). The majority of the patients developed severe COVID-19 pneumonia (22/40, 55%). After ED admission, COVID-19 pneumonia was complicated by ARDS in 10/40 (25%) patients. CPAP was required in 6/40 (15%) patients, while 10/40 (25%) patients underwent tracheal intubation. ICU admission was recorded in 11/40 (27%) patients.

In Table 1, results regarding PFTs at follow-up are also summarized. The median time elapsed between PFTs and FU CT was 2 days (IQR, 0–5 days). The median absolute FVC was 3.73 L (IQR, 3.06–4.46 L), corresponding to a median FVC % predicted of 102% (IQR, 87–112%). The median absolute FEV1 was 2.91 L (IQR, 2.25–3.64 L), with a median FEV1% predicted of 102% (IQR, 87–116%). The median FEV1/FVC ratio was 80% (IQR, 74–85%). %DLCO was available for 31/40 patients (77%) with a median value of 80% (IQR, 67–87%); %DLCO < 75% was identified in 11/31 (35%) patients.

### 3.2. CT Assessment

Table 2 shows data regarding both baseline and FU CT evaluations. The most frequent CT pattern at baseline was mixed ground glass and consolidations (22/40, 55%), followed by ground glass alone (14/40, 35%) and consolidations alone (4/40, 10%). At baseline, the median visual COVID-19 pneumonia extent was 25% (IQR, 20–40%), corresponding to a visual WAL extent of 72% (IQR, 60–80%). Median %m-HAA at admission was 32% (IQR, 20–54%), median %S-WAL was 66% (IQR, 46–77%), and median %LAAs was 0.15% (IQR, 0–0.75%). The median time elapsed between baseline CT and FU CT was 12 months (IQR, 8–23 months). At FU CT, the majority of the patients developed signs of fibrosis (22/40 patients, 55%), and most of the cases showed the appearance of bronchiectasis (16/40, 40%), followed by new architectural distortion (5/40, 12%) or honeycombing (1/40, 2%). Both median visual WAL extent (90%, IQR 85–100%; *p* < 0.0001) and %S-WAL (81%, IQR 73–86%; *p* < 0.0001) were significantly higher at follow-up CT. Conversely, there was a significant reduction in both median visual COVID-19 pneumonia extent (5%, IQR 0–10%; *p* < 0.0001) and %m-HAA (14%, IQR 11–19%; *p* < 0.0001) at FU CT. Median %LAAs was not significantly different at FU CT as compared to baseline CT (0.1%, IQR 0–1.75%; *p* = 0.463). Emphysema assessed visually was identified in 7/40 (17%) patients, both at baseline and follow-up CT.

### 3.3. Agreement Analysis at FU CT

The agreement between readers for visual assessment of both %WAL (ICC 0.889, 95% CIs 0.791–0.941) and COVID-19 pneumonia extent (ICC 0.950, 95% CIs 0.907–0.974) was excellent. Moderate agreement was found between the visual and software assessments of both %WAL (ICC 0.616, 95% CIs 0.274–0.796) and %m-HAA (ICC 0.590, 95% CIs 0.225–0.783). A poor (Figure 2) agreement (Kw 0.184; 95% CIs −0.098–0.623) was identified between the visual assessment of emphysema and the software assessment of %LAAs ≥ 5%. 

### 3.4. Correlation Analysis at FU CT

The rho correlation coefficients between quantitative CT parameters and PFTs at follow-up CT are summarized in Table 3. The correlation between visual %WAL, software %WAL, visual COVID-19 pneumonia extent, %m-HAA, and all PFTs was not statistically significant (*p* > 0.05). The software quantification of total lung volume showed a significant, positive, correlation with absolute FVC (rho 0.515, 95% CIs 0.243–0.712, *p* = 0.0007) and with absolute FEV1 (rho 0.362, 95% CIs 0.057–0.605, *p* = 0.021); the correlation between software quantification of total lung volume and the remaining PFTs was not significant (*p* > 0.05). In addition, a significant, negative, correlation was identified between %LAAs and both FEV1/FVC ratio (rho −0.410, 95% CIs −0.639–−0.112, *p* = 0.008) and %DLCO (rho −0.426, 95% CIs −0.678–−0.084, *p* = 0.017). The correlation between %LAAs and the remaining PFTs was not significant (*p* > 0.05).

Although not significant, a trend towards a higher rate of patients with %DLCO < 75% was identified in patients with emphysema detected visually (75% vs. 25%, *p* = 0.115) and with %LAA ≥ 5% (57% vs. 43%, *p* = 0.209).

## 4. Discussion

The aim of the present study was to test the correlation between PFTs and quantitative parameters obtained at FU CT performed at least six months after COVID-19 pneumonia onset. Between baseline and follow-up CT, a significant reduction in COVID-19 pneumonia was assessed both visually and by software, while the %LAA was similar. Among all parameters quantified at follow-up CT, a significant positive correlation was identified between software measurements of lung volume and both FVC and FEV1. In addition, a significant inverse correlation was demonstrated between %LAAs and both %DLCO and FEV1/FVC ratio. 

According to previous studies, we identified a significant reduction in the extent of pulmonary abnormalities at follow-up CT, assessed both visually and by software [5,11]. Han et al. explored CT scans of 144 patients affected by COVID-19 pneumonia, demonstrating a significant reduction in the extent of pulmonary abnormalities at follow-up CTs at 6, 12, and 24 months assessed visually by a semiquantitative score [5]. Barini et al., using commercially available software, reported a reduction of around 15% of whole lung volume pulmonary abnormalities at follow-up CT performed 18 months after diagnosis [11].

Furthermore, among parameters defined at quantitative CT, only %LAAs was significantly associated with %DLCO. The relationship between %DLCO and CT abnormalities or quantitative CT parameters at follow-up of patients affected by COVID-19 pneumonia is controversial [5,11]. Two years after COVID-19 pneumonia diagnosis, patients with residual interstitial lung abnormalities (ILAs) at CT more frequently manifested %DLCO of ≤75% [5]. Conversely, in a recent study, no significant relationship was found between PFTs and quantitative CT parameters, namely, healthy parenchyma, ground glass opacities, consolidations, and pulmonary volume [11]. 

New emphysematous abnormalities were identified visually in around one quarter of the patients with COVID-19 pneumonia mechanically ventilated 3 months after hospital discharge [22]. These changes were noted both in infiltrated areas and outside them at CT and were considered, respectively, sequelae of direct parenchymal destruction caused by infection and ventilator-induced injury [22]. Faverio et al. visually detected emphysema at FU CT in 11% of COVID-19 patients 12 months after infection; emphysema was more frequent in patients who underwent CPAP or mechanical ventilation [23]. Modifications in LAAs between baseline CT and follow-up CT were explored in 32 patients with COVID-19 pneumonia [12]. In follow-up CT performed >30 days after diagnosis, Celik et al. reported a median LAAs around 150cc higher as an absolute value, corresponding to around 5% increase relative to whole lung volume [12]. By contrast, we identified lower %LAAs (around 0.1% in our study vs. 13% at follow-up CT detected by Celik et al.), and we failed to demonstrate an increase in %LAAs at follow-up CT. This discrepancy could be explained by the different software analysis, considering that we applied a noise reduction algorithm; in addition, the rate of COPD patients in our sample was only 7%, which was not specified by Celik et al. [12]. Nevertheless, in our study, %LAAs were significantly associated with %DLCO. Emphysema development is a consequence of airway damage distal to the terminal bronchioles, determined by noxious agents (e.g., cigarette smoke) or infection [24]. SARS-CoV-2 infects cells of the bronchial mucosa by binding the angiotensin converting enzyme (ACE) 2 receptor, leading to surfactant loss and edema, with a consequent tendency for the small airway to collapse [25]. Autopsy studies in COVID-19 pneumonia revealed the frequent presence of small airway inflammation with alveolar hyaline membranes and type 2 pneumocyte hyperplasia [26]. After the COVID-19 infection, mid- and long-term changes both intraluminal and in the surrounding parenchyma determine remodeling of the small airways [25]. Air trapping was detected at CT in around 35% of the patients with persistent symptoms after COVID-19 infection, regardless of the severity of the acute process and even a year after the onset of the disease [25,27]. Thus, we speculate that pulmonary inflammation could lead to emphysematous changes associated with reduced lung function. Additionally, as previously reported, we demonstrated poor agreement between visual assessment of emphysema and %LAAs even in follow-up CT of COVID-19 pneumonia [28]. In several patients, after the clearance of the virus, symptoms persist or worsen, with only a mild abnormality detected visually on a chest CT [1,2]. Thus, some authors raised the concern that some key findings are missing and may be revealed by high-quality CT, which is nearly universally available and can provide evidence of new lung findings [2]. For these reasons, the software quantification of %LAAs could be a surrogate marker of irreversible damage and reduced lung function in the follow-up of patients who suffered from COVID-19 pneumonia. Our results, despite being obtained on a small number of patients, suggest performing a quantification of %LAAs in patients with persistent symptoms and\or impaired %DLCO to identify changes that can be hardly identified visually.

The present study has several limitations. 1. It is a retrospective study from a single institution on a small number of patients. However, we attempt to obtain a homogeneous sample in terms of follow-up CT interval (>6 months) and PFTs (within 6 months from follow-up CT). 2. This study included patients selected during the first pandemic wave in a prevaccination era; COVID-19 variants and vaccination status could lead to different lung damage patterns and relative symptoms. 3. Suboptimal inspiration can confound the quantification of lung volumes and LAAs by software. Nevertheless, the study protocol tried to reduce this limitation by coaching the patients to obtain a deep, full inspiration prior to CT scans. 4. Quantification by software of LAAs could differ on the basis of different scanners, kernels, and software versions; since CT scans were acquired with different scanners and kernels, a noise-reduction algorithm was used considering that it demonstrated a better correlation with PFTs [29]. 5. An expiratory scan was not performed at follow-up CT, thus “air trapping” could not be evaluated, but areas lower than −950 HU at the inspiratory CT scan are universally considered areas of emphysema [19]. 6. PFTs of longitudinal changes are not available considering that spirometry was not performed at baseline due to the severity of the symptoms (55% of the patients manifested severe COVID-19 pneumonia at ED admission). 7. Since all patients were alive at follow-up CT, no histological sample to confirm the presence of emphysema identified at CT was available.

## 5. Conclusions

At FU CT performed at least 6 months after COVID-19 pneumonia onset, %LAAs quantified by software were significantly, inversely, associated with %DLCO. The quantification of %LAAs could be a marker of impaired lung function, helping to identify patients with COVID-19 pneumonia sequelae despite subtle changes at visual evaluation of lung CT.

## Figures and Tables

**Figure 1 diagnostics-13-03328-f001:**
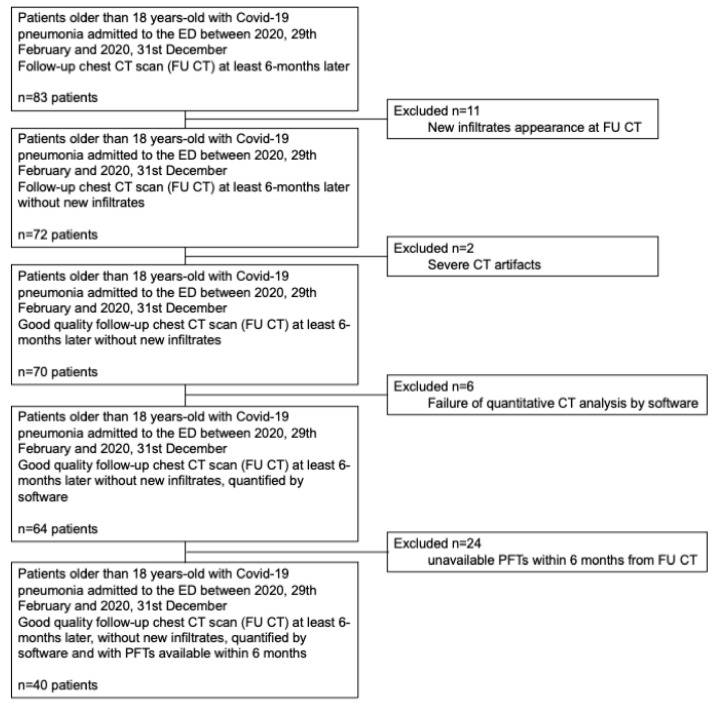
The diagram shows the patient selection process. Abbreviations: CT, computed tomography; ED, emergency department; FU, follow-up; PFTs, pulmonary function tests.

**Figure 2 diagnostics-13-03328-f002:**
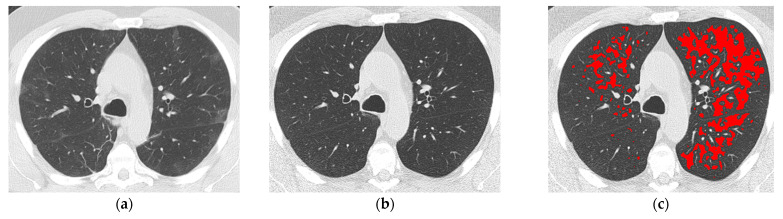
A 55-year-old never-smoker man with COVID-19 pneumonia detected in March 2020. The baseline axial high-resolution computed tomography (HRCT) image at the level of the aortic arch (**a**) was typical for COVID-19 pneumonia and characterized mainly by bilateral ground-glass opacities, which extended to 20% of the whole lung parenchyma (visual score). The axial follow-up HRCT image at the same level performed 30 months later (**b**) identified complete resolution of COVID-19 pneumonia without emphysema at visual assessment. (**c**) The same axial HRCT in (**b**) with highlighted in red low attenuation areas (LAAs) identified by software analysis showed 20% %LAAs extent relative to the whole lung volume; pulmonary function tests performed 7 days before follow-up HRCT showed a reduced %DLCO (49%).

**Table 1 diagnostics-13-03328-t001:** Patients’ demographics, comorbidities, COVID-19 pneumonia details, and PFTs values at FU CT.

Variables	All Patients (*n* = 40)
Age (years)	63 (56–71)
Gender (n)	
Males	25/40 (62%)
Females	15/40 (38%)
Current or former smoker (*n*)	6/40 (15%)
Pulmonary comorbidities (*n*)	
COPD	3/40 (7%)
Asthma	2/40 (5%)
OSAS	2/40 (5%)
Cardiovascular comorbidity (*n*)	11/40 (28%)
Hypertension (*n*)	18/40 (45%)
Diabetes (*n*)	3/40 (7%)
Neurological comorbidity (*n*)	2/40 (5%)
Oncological comorbidity (*n*)	5/40 (8%)
Respiratory rate at admission (breaths/min)	20 (18–25)
Blood oxygen saturation at admission (%)	92 (89–96)
Severe COVID-19 at admission (*n*)	22/40 (55%)
COVID-19 complicated by ARDS (*n*)	10/40 (25%)
CPAP (*n*)	6/40 (15%)
ICU admission (*n*)	11/40 (27%)
Tracheal intubation (*n*)	10/40 (25%)
FVC at follow-up CT (L)	3.73 (3.06–4.46)
FVC predicted at follow-up CT (%)	102 (87–112)
FEV1 at follow-up CT (L)	2.91 (2.25–3.64)
FEV1 predicted at follow-up CT (%)	102 (87–116)
FEV1/FVC ratio at follow-up CT (%)	80 (74–85)
DLCO at follow-up CT (%) ^1^	80 (67–87)
%DLCO < 75% at follow-up CT (*n*) ^1^	11 (35%)

Data are shown as counts and percentages in brackets for categorical variables or median and interquartile range in brackets for continuous variables. ^1^ Data are available for 31 patients. Abbreviations: ARDS, acute respiratory distress syndrome; COPD, chronic obstructive pulmonary disease; CPAP, continuous positive airway pressure; CT, computed tomography; DLCO, diffusion lung carbon monoxide; FEV1, forced expiratory volume in the first second; FVC, forced vital capacity; ICU, intensive care unit; OSAS, obstructive sleep apnea syndrome.

**Table 2 diagnostics-13-03328-t002:** Quantitative CT parameters assessed at baseline and at follow-up (*n* = 40).

Quantitative CT Parameters	Baseline CT	Follow-Up CT	*p*-Value
Total lung volume (L)	4.14 (3.51–5.53)	5.57 (4.59–6.42)	<0.0001
Visual WAL extent (%)	72 (60–80)	90 (85–100)	<0.0001
Software WAL extent (%)	66 (46–77)	81 (73–86)	<0.0001
Visual pneumonia extent (%)	25 (20–40)	5 (0–10)	<0.0001
HAAs > −750 HU extent (%)	32 (20–54)	14 (11–19)	<0.0001
LAAs < −950 HU extent (%)	0.15 (0–0.75)	0.1 (0–1.75)	0.463

Abbreviations: CT, computed tomography; HAAs, high attenuation areas; HU, Hounsfield units; LAAs, low attenuation areas; WAL, well-aerated lung.

**Table 3 diagnostics-13-03328-t003:** Spearman’s correlations between quantitative CT parameters and pulmonary function tests at follow-up after COVID-19 pneumonia.

PFTs	Total Lung Volume (L)	Visual WAL (%)	Visual Pneumonia Extent FU (%)	Software WAL (%)	HAAs −750 HU (%)	LAAs −950 HU (%)
FVC (L)	**0.515 (0.243;0.712)**	0.216 (−0.102;0.494)	0.054 (−0.261;0.360)	0.156 (−0.164;0.446)	−0.175 (−0.461;0.144)	0.117 (−0.202;0.413)
FVC (% predicted)	0.164 (−0.156;0.452)	0.134 (−0.185;0.428)	−0.042 (−0.350;0.272)	0.221 (−0.096;0.498)	−0.260 (−0.528;0.056)	0.200 (−0.119;0.482)
FEV1 (L)	**0.362 (0.057;0.605)**	0.203 (−0.116;0.484)	0.169 (−0.150;0.457)	0.139 (−0.180;0.432)	−0.052 (−0.358;0.264)	−0.062 (−0.367;0.254)
FEV1 (% predicted)	0.028 (−0.285;0.337)	0.095 (−0.223;0.395)	0.181 (−0.138;0.466)	0.149 (−0.170;0.440)	−0.022 (−0.331;0.291)	−0.045 (−0.352;0.269)
FEV1/FVC ratio (%)	−0.173 (−0.460;0.146)	−0.051 (−0.357;0.264)	0.292 (−0.021;0.554)	0.013 (−0.300;0.323)	0.260 (−0.055;0.529)	**−0.410 (−0.639;−0.112)**
DLCO (%)	0.261 (−0.103;0.563)	0.283 (−0.079;0.579)	−0.015 (−0.368;0.341)	0.212 (−0.154;0.527)	0.127 (−0.238;0.461)	**−0.426 (−0.678;−0.084)**

Values correspond to rho Spearman’s coefficient of rank correlation, with 95% confidence intervals in brackets. Values in bold are statistically significant (*p* < 0.05). Abbreviations: DLCO, diffusion lung carbon monoxide; FEV1, forced expiratory volume in the first second; FVC, forced vital capacity; HAAs, high attenuation areas; HU, Hounsfield units; LAAs, low attenuation areas; WAL, well-aerated lung.

## Data Availability

The data presented in this study are available on request from the corresponding author. The data are not publicly available due to privacy policy restrictions.

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
