# Peer review of "Quantitative CT at Follow-Up of COVID-19 Pneumonia: Relationship with Pulmonary Function Tests"

_diagnostics, 2023, doi:10.3390/diagnostics13213328_

Round 1

Reviewer 1 Report

Comments and Suggestions for Authors

The authors present an interesting study, trying to correlate post-COVID-19 PFT with quantitative and visual CT parameters. Among a group of mostly severe-critical COVID-19 survivors, they identified CT changes suggestive of emphysema (LAA), which correlated FEV1/FVC ratio and DLCO. Those results are interesting, yet major revision of the manuscript is required to confer it more accessible to readers.

some of my major concerns include:

1. English need significant editing and improvement

2.The study methodology must be better explained: how exactly were subjects selected for the study? where they all symptomatic at FU and recruited from a FU-clinic? otherwise, why were FU CT and PFT performed (what were the indications)? how was the sample size calculated (if it was calculated)? what equations were used for %pred of PFT, and why didn't the authors used Z-statistics as suggested by current guidelines for PFT interpretation? The maximal severity of COVID-19 should also be reported, as well as more comorbidities (obesity, diabetes, heart disease etc.). Also, Tiffenau index is a rarely used term, please use FEV1/FVC ratio. 

I don't understand how consent was obtained for a retrospective study. This has to do with the study design about subjects recruitment which is unclear (which subjects were chosen to undergo FU CT and PFT and why...)

3. in the results section some more information is needed, regarding time from FU CT to PFT, more details on subjects as in point 2. was there an association between disease severity during COVID-19 and outcomes?

4. Please present tables differently: show data of first CT and FU CT side-by-side for comparison. perhaps in another table.

5. The discussion section need reinforcements. You should discuss mechanisms more: does changes relate to COVID-19 itself? to damage from mechanical ventilation? to prior undiagnosed changes on CT? There are studies contemplating small airway disease after COVID-19 which may be relevant. compare results to recent data on long term CT after COVID-19 - many studies show only minimal changes, is it because patient selection?

6. Add more info on the study limitations, including technichs to assess emphysema/LAA, selection bias, high drop-out rates, viral variants during the study period and lack of vaccination at that time, and how that changed in contemporary COVID-19 waves, etc. 

7. maybe add a paragraph about possible clinical implications of the study, is any

8. update references list according to Discussion

Comments on the Quality of English Language

Overall needs editing to improve coherance.  

Author Response

Reviewer 1

The authors present an interesting study, trying to correlate post-COVID-19 PFT with quantitative and visual CT parameters. Among a group of mostly severe-critical COVID-19 survivors, they identified CT changes suggestive of emphysema (LAA), which correlated FEV1/FVC ratio and DLCO. Those results are interesting, yet major revision of the manuscript is required to confer it more accessible to readers.

some of my major concerns include:

  1. English need significant editing and improvement.

Response: thank you. We edited the language as suggested.

2.The study methodology must be better explained: how exactly were subjects selected for the study? where they all symptomatic at FU and recruited from a FU-clinic? otherwise, why were FU CT and PFT performed (what were the indications)? how was the sample size calculated (if it was calculated)? what equations were used for %pred of PFT, and why didn't the authors used Z-statistics as suggested by current guidelines for PFT interpretation? The maximal severity of COVID-19 should also be reported, as well as more comorbidities (obesity, diabetes, heart disease etc.). Also, Tiffenau index is a rarely used term, please use FEV1/FVC ratio.

I don't understand how consent was obtained for a retrospective study. This has to do with the study design about subjects recruitment which is unclear (which subjects were chosen to undergo FU CT and PFT and why...)

Response: thank you for these important remark.

We have specified that patients underwent follow-up CT and PFTs in case of more severe clinical disease course, or in patients presenting with new or progressive respiratory symptoms in the mid-long-term after acute Covid-19 pneumonia [page 2, lines 90-92: “FU CT was performed in more severe clinical disease course, or in patients presenting with new or progressive respiratory symptoms in the mid-long-term after acute Covid-19 pneumonia”].

Since it was a retrospective study from a single center, the sample size was not calculated, we include all patients during the period that fulfill inclusion and exclusion criteria.

Considering that patients selection started in 2020, we used the previous guidelines for the interpretation of PFTs (Eur Respir J 2005; 26: 948–968); we have specified it within the text [page 3, lines 127-129: “Interpretation of the PFTs was based on the American Thoracic Society (ATS) and ERS guidelines at the time of patients selection [15]”].

Covid-19 severity was reported in table 1 and underlined in the Results section, 55% of the patients manifested severe COVID-19 pneumonia at admission and 25% were complicated by ARDS [page 5, lines 210-211: “The majority of the patients developed severe Covid-19 pneumonia (22/40, 55%). After ED admission Covid-19 pneumonia was complicated by ARDS in 10/40 (25%) patients”].

We have reported additional comorbidities frequencies in table 1 and in the Results section categorizing them as cardiovascular, hypertension, diabetes, neurological, and oncological [page 5, lines 206-208: “Beside pulmonary comorbidities, the majority of the patients was affected by systemic hypertension (18/40, 45%)”.]

We have substituted Tiffenau index term with FEV1/FVC ratio as suggested throughout the whole text.

Informed consent was obtained by all subjects included in the study as better specified in Methods [page 2, lines 85-86: “Informed consent was obtained from all subjects involved in the study”].

  1. in the results section some more information is needed, regarding time from FU CT to PFT, more details on subjects as in point 2. was there an association between disease severity during COVID-19 and outcomes?

Response: thank you for these important suggestions.

As specified in Methods, patients without PFTs available within 6 months from FU CT were excluded from the study [page 2, lines 92-94: “Exclusion criteria were: ….4. unavailable PFTs within 6 months from FU CT”].

We have added patients comorbidities (see point 2) and median time elapsed between PFTs and FU CT in the results section and in table 1 [page 5, lines 206-208: “Beside pulmonary comorbidities, the majority of the patients was affected by systemic hypertension (18/40, 45%)”; page 5, lines 214-215: “The median time elapsed between PFTs and FU CT was 2 days (IQR, 0-5 days)” ].

The aim of our study was to test the association between quantitative parameters assessed by CT and PFTs at follow-up, thus we did not explore the eventual association between Covid-19 severity and outcome.

  1. Please present tables differently: show data of first CT and FU CT side-by-side for comparison. perhaps in another table.

Response: thank you for this suggestion. We have added Table 2 showing data of first CT and FU CT side-by-side for comparison.

  1. The discussion section need reinforcements. You should discuss mechanisms more: does changes relate to COVID-19 itself? to damage from mechanical ventilation? to prior undiagnosed changes on CT? There are studies contemplating small airway disease after COVID-19 which may be relevant. compare results to recent data on long term CT after COVID-19 - many studies show only minimal changes, is it because patient selection?

Response: thank for these insightful suggestions.

We have added more data regarding emphysema development and small airways damage in Covid-19. Particularly, we have cited further articles that reported emphysema development after Covid-19 pneumonia, related both to mechanical ventilation and infection damage; in addition we have underlined that around 30% of the patients that manifested persistent symptoms after Covid-19 infection showed air-trapping at FU CT, regardless severity of the acute infection [pages 8-9, lines 359-374: “New emphysematous abnormalities were identified visually in around one-quarter of the patients with Covid-19 pneumonia mechanically ventilated 3 months after hospital discharge [22]. These changes were noted at CT both in infiltrated areas and outside them, considered respectively sequalae of direct parenchymal destruction caused by infection and ventilator-induced injury [22]. Faverio et al. detected visually emphysema at FU CT in 11% of Covid-19 patients 12 months after infection; emphysema was more frequent in patients who underwent CPAP or mechanical ventilation”; page 9, lines 386-395: “SARS-CoV-2 infects cells of bronchial mucosa by binding the angiotensin converting enzyme (ACE) 2 receptor, leading to surfactant loss and edema with consequent tendence of the small airways to collapse [25]…After the Covid-19 infection, mid and long-term changes both intraluminal and of the surrounding parenchyma determine remodeling of the small airways [25]. Air-trapping was detected at CT in around 35% of the patients with persistent symptoms after Covid-19 infection, regardless the severity of the acute process and even a year after onset of the disease [25,27]”].

  1. Add more info on the study limitations, including technichs to assess emphysema/LAA, selection bias, high drop-out rates, viral variants during the study period and lack of vaccination at that time, and how that changed in contemporary COVID-19 waves, etc.

Response: thank you for this important remark.

We have specified further limitations of the study regarding patient selection during the first pandemic wave (viral variants, vaccination status) and technical difficulties for the quantification of the LAAs [page 9, lines 411-413:” This study included patients selected during the first pandemic wave in a pre-vaccination era; Covid-19 variants and vaccination status  could lead to different lung damage pattern and relative symptoms”; page 9, lines 416-419: “Quantification by software of LAAs could differ on the basis of different scanner, kernel, and software versions; since CT scans were acquired with different scanner and kernel, a noise-reduction algorithm was used considering that it was demonstrated a better correlation with PFTs [29].”]

  1. maybe add a paragraph about possible clinical implications of the study, is any

Response: thank you for this suggestion.

We have added a short possible clinical implications [page 9, lines 404-407: “Our results, despite obtained on a small number of patients, suggest to perform a quantification of %LAAs in patients with persistent symptoms and\or impaired %DLCO to identify changes, that can be hardly identified visually.”].

  1. update references list according to Discussion

Response: as a consequence of the Discussion reinforcement we have added reference number 22, 23, 25, and 27.

Reviewer 2 Report

Comments and Suggestions for Authors

Dear authors, 

The article is well-balanced. Although some of the terms you used are known, you should use the descriptive form for the first time you introduce it (as LAAs in the abstract).   

Author Response

Dear authors,

The article is well-balanced. Although some of the terms you used are known, you should use the descriptive form for the first time you introduce it (as LAAs in the abstract).

Response: thank you.

We have used the descriptive form of the abbreviations the first time we mentioned them as suggested.

Reviewer 3 Report

Comments and Suggestions for Authors

The objectives of the paper should clearly explained.

The abstract is not well written... It has to show part of the objective, method, results and implications of the study. it is important to reframe it to reflect the questions addressed in the paper. 

Test bed and parameters for assessment not furnished in abstract. 

Authors stated that the study included 40 patients. Why do you limited with less number of patients?

The conclusions should be given in more compherensive manner.

The recent published papers not cited and referred in literature survey.

2.3. Statistical Analysis- Have you used any data sets for this analysis?

Line no 201- % predicted of 102% (IQR, 87-116%)- Is it correct?

Author Response

Reviewer 3

  1. The objectives of the paper should clearly explained.

Response: the aim of the study was underlined at the end of the introduction [page 2, lines 78-80: “the aim of the present study was to test the association between PFTs and quantitative parameters assessed visually or by software at FU CT scans at least 6-months after Covid-19 pneumonia onset”].

  1. The abstract is not well written... It has to show part of the objective, method, results and implications of the study. it is important to reframe it to reflect the questions addressed in the paper.

Response: thank you.

We have reframed the abstract to reflect questions addressed in the paper.

  1. Test bed and parameters for assessment not furnished in abstract.

Response: thank you.

We have specified the quantitative parameters assessed at FU CT in abstract [page 1, lines 20-22 “At FU-CT were identified quantitative parameters of well-areated lung and pneumonia extent both visually and by software, using CT density thresholds”; page 1, lines 25-26: “A significant correlation was identified between low attenuation areas% (%LAAs) <950 Hounsfield units (HU) and both….”].

The study sample was constituted by 40 patients, 38% females, with a median age 63 years-old (IQR, 56-71 years-old) as specified at the beginning of the results section (page 1, lines 24-25).

  1. Authors stated that the study included 40 patients. Why do you limited with less number of patients?

Response: we agree that the number of patients in small. Nevertheless, the study is retrospective from a single center, and in order to have an homogeneous sample many patients from the initial cohort were excluded. We have underlined this limitation in the Discussion [page 9, lines 408-411: “The present study has several limitations. 1. It is a retrospective study, from a single institution on a small number of patients. However, we attempt to obtain an homogeneous sample in terms of follow-up CT interval (>6 months) and PFTs (within 6 months from follow-up CT).”]

  1. The conclusions should be given in more compherensive manner.

Response: thank you for this suggestion.

We have rephrased the Conclusion paragraph [page 10, lines 446-450: “At FU CT performed at least 6 months after Covid-19 pneumonia onset, %LAAs quantified by software were significantly, inversely, associated with %DLCO. The quantification of %LAAs could be a marker of the impaired lung function, helping to identify patients with Covid-19 pneumonia sequelae despite subtle changes at visual evaluation of lung CT.”]

  1. The recent published papers not cited and referred in literature survey.

Response: thank you for this suggestion. As suggested even by Reviewer 1 we have reinforced the Discussion and we have added reference number 22, 23, 25, and 27.

  1. Statistical Analysis- Have you used any data sets for this analysis?

Response: yes we used a dataset. We have specified it in Methods [page 5, lines 198-200: “All data were recorded using a dedicated database (Excel 2010, Microsoft Corp., Red-mond, WA), and statistical analysis was performed using MedCalc software (version 14.8.1, MedCalc Software Ltd, Ostend, Belgium).”]

  1. Line no 201- % predicted of 102% (IQR, 87-116%)- Is it correct?

Response: we have checked data, it was correct.

Round 2

Reviewer 1 Report

Comments and Suggestions for Authors

The authors have made significant efforts to improve the manuscript, and this should be recognized.

One interesting topic that might be added to the discussion and the lack of correlation between CT findings and FEV1/FVC ratio. One might suspect that emphysematous changes and air-trapping will be reflected in an obstructive PFT pattern (FEV1/FVC<0.7, or below normal), but that is not the case. If the authors have available values of medium-small size airways expiratory flows (MEF25-75% etc.) to calculate the correlation with CT abnormalities it could be interesting. otherwise, please mention that small airway disease may not be apparent in standard PFT.

Comments on the Quality of English Language

slight revision by an English speaker may improve manuscript

Reviewer 3 Report

Comments and Suggestions for Authors

It has been improved as per comments. Now the potential contribution and visibility improved.